# Relationship between circulating FSH levels and body composition and bone health in patients with prostate cancer who undergo androgen deprivation therapy: The BLADE study

Marco Bergamini[1], Alberto Dalla Volta[1]*, Carlotta Palumbo[2], Stefania Zamboni[3], Luca Triggiani[4], Manuel Zamparini[1], Marta Laganà[1], Luca Rinaudo[5], Nunzia Di Meo[6], Irene Caramella[1], Roberto Bresciani[7], Francesca Valcamonico[1], Paolo Borghetti[4], Andrea Guerini[4], Davide Farina[6], Alessandro Antonelli[8], Claudio Simeone[3], Gherardo Mazziotti[9,10]*, Alfredo Berruti[1]

[1]Medical Oncology Unit, ASST Spedali Civili, Department of Medical and Surgical Specialties, Radiological Sciences and Public Health, University of Brescia, Brescia, Italy; [2]Division of Urology, Department of Translational Medicine, University of Eastern Piedmont, Maggiore Della Carità Hospital, Novara, Italy; [3]Urology Unit, ASST Spedali Civili, Department of Medical and Surgical Specialties, Radiological Sciences and Public Health, University of Brescia, Brescia, Italy; [4]Radiation Oncology Unit, ASST Spedali Civili, Department of Medical and Surgical Specialties, Radiological Sciences and Public Health, University of Brescia, Brescia, Italy; [5]Tecnologie Avanzate S.r.l, Turin, Italy; [6]Radiology Unit, ASST Spedali Civili, Department of Medical and Surgical Specialties, Radiological Sciences and Public Health, University of Brescia, Brescia, Italy; [7]Division of Biotechnology, Department of Molecular and Translational Medicine (DMTM), University of Brescia, Brescia, Italy; [8]Urology Unit, AOUI Verona, Department of Surgery, Dentistry, Pediatrics and Gynecology, University of Verona, Verona, Italy; [9]Department of Biomedical Sciences, Humanitas University, Pieve Emanuele-Milan, Milan, Italy; [10]Endocrinology, Diabetology and Medical Andrology Unit, Metabolic Bone Diseases and Osteoporosis Section, IRCCS Humanitas Research Hospital,, Milan, Italy

*For correspondence:
alberto.dallavolta@gmail.com (ADV);
gherardo.mazziotti@hunimed.eu (GM)

## Abstract

**Background:** Among its extragonadal effects, follicle-stimulating hormone (FSH) has an impact on body composition and bone metabolism. Since androgen deprivation therapy (ADT) has a profound impact on circulating FSH concentrations, this hormone could potentially be implicated in the changes of fat body mass (FBM), lean body mass (LBM), and bone fragility induced by ADT. The objective of this study is to correlate FSH serum levels with body composition parameters, bone mineral density (BMD), and bone turnover markers at baseline conditions and after 12 months of ADT.

**Methods:** Twenty-nine consecutive non-metastatic prostate cancer (PC) patients were enrolled from 2017 to 2019 in a phase IV study. All patients underwent administration of the luteinizing hormone-releasing hormone antagonist degarelix. FBM, LBM, and BMD were evaluated by dual-energy x-ray absorptiometry at baseline and after 12 months of ADT. FSH, alkaline phosphatase, and C-terminal

telopeptide of type I collagen were assessed at baseline and after 6 and 12 months. For outcome measurements and statistical analysis, *t*-test or sign test and Pearson or Spearman tests for continuous variables were used when indicated.

**Results:** At baseline conditions, a weak, non-significant, direct relationship was found between FSH serum levels and FBM at arms ($r = 0.36$) and legs ($r = 0.33$). Conversely, a stronger correlation was observed between FSH and total FBM ($r = 0.52$, $p = 0.006$), fat mass at arms ($r = 0.54$, $p = 0.004$), and fat mass at trunk ($r = 0.45$, $p = 0.018$) assessed after 12 months. On the other hand, an inverse relationship between serum FSH and appendicular lean mass index/FBM ratio was observed ($r = -0.64$, $p = 0.001$). This is an ancillary study of a prospective trial and this is the main limitation.

**Conclusions:** FSH serum levels after ADT could have an impact on body composition, in particular on FBM. Therefore, FSH could be a promising marker to monitor the risk of sarcopenic obesity and to guide the clinicians in the tailored evaluation of body composition in PC patients undergoing ADT.

**Funding:** This research was partially funded by Ferring Pharmaceuticals. The funder had no role in design and conduct of the study, collection, management, analysis, and interpretation of the data and in preparation, review, or approval of the manuscript.

**Clinical trial number:** clinicalTrials.gov NCT03202381, EudraCT Number 2016-004210-10.

## Editor's evaluation

This ancillary study from the BLADE study provided valuable findings regarding FSH as a predictor of sarcopenic obesity in patients using the LHRH antagonist Degarelix, suggesting a potential direct effect of FSH on fat. The study design and analysis were solid, and the limitations of the study such as the small sample size and lack of sex hormone analysis-were properly addressed.

## Introduction

Recent preclinical and clinical studies have shown that follicle-stimulating hormone (FSH) exerts effects beyond those on gonadal tissue, which have been known for a long time (*Lizneva et al., 2019*). In the animal model, FSH has been observed to stimulate osteoclastic activity and therefore to exert a negative role on bone mass (*Sun et al., 2006*; *Zaidi et al., 2023*; *Gera et al., 2022*). Furthermore, FSH was found to have a positive action on adipocytes and the blockade of FSH induced by antibodies against FSH receptor caused an increase in bone mass and a reduction of body fat (*Liu et al., 2017*; *Rojekar et al., 2023*). These preclinical data were confirmed in clinical studies. Large epidemiologic data have shown significant reductions in bone mineral density (BMD) and high resorption rates 2–3 years prior to menopause, when FSH serum levels are increasing, which is also associated with increased body weight and visceral adiposity (*Thurston et al., 2009*; *Senapati et al., 2014*). A longitudinal study involving post-menopausal women has shown that increases in circulating FSH levels were associated with greater increases in the percentage of total body fat, total body fat mass, and subcutaneous adipose tissue (*Mattick et al., 2022*). Moreover, clinical studies have found that FSH is also involved in the modulation of lean mass. A large cohort study of perimenopausal women showed that increased FSH concentrations were associated with increasing fat mass and decreasing lean mass measured by bioelectrical impedance analysis (*Sowers et al., 2007*). Another recent prospective study showed that the indicators of sarcopenia were strongly associated with gonadotropins levels, especially in older men (*Guligowska et al., 2021*). The impact of FSH on body composition may at least partially explain the observed correlation between FSH levels and the risk of cardiovascular events (*Munir et al., 2012*; *Silva et al., 2013*; *El Khoudary et al., 2016*).

Studies clarifying the relationship among FSH, body composition measures, and BMD in prostate cancer (PC) patients are lacking. Likewise, the interaction between the variation of FSH serum levels and the modification of these parameters in PC patients undergoing androgen deprivation therapy (ADT) is not known.

We recently conducted the BLADE study (Bone mineraL mAss Dexa dEgarelix), a prospective phase IV study designed to obtain explorative information on dual-energy x-ray absorptiometry (DXA) measurement changes in lean body mass (LBM) and fat body mass (FBM) in patients with non-metastatic PC treated with degarelix, an LHRH (luteinizing hormone-releasing hormone) antagonist

**eLife digest** Treatments given to cancer patients can cause negative side effects. For example, a treatment known as androgen deprivation therapy – which is used to reduce male sex hormone levels in prostate cancer patients – can lead to increased body fat percentage and decreased bone density. These adverse effects can have further negative impacts on patient health, such as increasing the risk of cardiovascular disease and fractures from falls from standing height or less, respectively. Understanding how androgen deprivation therapy contributes to these negative side effects may help clinicians better manage care and outcomes for patients with prostate cancer.

Follicle stimulating hormone (or FSH for short) has roles in male and female reproduction but has also been linked to changes in body composition. For example, elevated FSH levels are associated with higher total fat body mass in post-menopausal women. While androgen deprivation therapy is known to alter FSH blood levels, the impact of this change in prostate cancer patients was not well understood.

To investigate the effect of androgen deprivation therapy on FSH levels and body composition, Bergamini et al. used X-ray technology to measure total fat body mass in prostate cancer patients before and after undergoing 12 months of androgen deprivation therapy. The findings showed that patient FSH blood levels significantly decreased after 12 months of treatment. Higher FSH blood levels strongly correlated with increased total fat body mass after 12 months of treatment.

The findings of this clinical trial suggest that FSH blood levels impact the body composition of patients undergoing androgen deprivation therapy. As a result, FSH blood levels may be a suitable biomarker for identifying patients that are more likely to develop obesity and are therefore at greater risk of complications such as cardiovascular disease.

that has been developed to achieve effective long-term medical castration without the risk of testosterone surge and its associated flare. Data on changes in total LBM and FBM and on the impact of degarelix on BMD and bone turnover markers were recently published (*Palumbo et al., 2021*; *Palumbo et al., 2022*).

Here, we analyse the association of FSH with DXA-assessed FBM, LBM, and BMD and bone turnover markers at baseline and after degarelix administration in the patients enrolled in the BLADE study.

## Methods

### Trial design and endpoints

BLADE is a single-centre, prospective, interventional phase IV study (clinicalTrials.gov NCT03202381, EudraCT Number 2016-004210-10) conducted at the Prostate Cancer Unit of the Azienda Socio Sanitaria Territoriale degli Spedali Civili and Università degli Studi of Brescia. The study was carried out in accordance with the Declaration of Helsinki Principles and Good Clinical Practices and was approved by the Ethics Committee of Brescia (approval number NP2540). All patients provided a written informed consent. Male patients with histologically confirmed PC without bone metastasis at bone scintigraphy, judged eligible to ADT according to current guidelines recommendations (*Cornford et al., 2017*; *Mottet et al., 2017*) after a multidisciplinary discussion, were enrolled (*Supplementary file 1*). Eligibility criteria have been published elsewhere (*Palumbo et al., 2021*). Degarelix was administered as a subcutaneous injection with a starting dose of 240 mg, followed by a maintenance dose of 80 mg every 28 days. After 12 months, treatment with degarelix was continued as clinically indicated. DXA measurements for assessing BMD and body composition parameters were performed at baseline and after 12 months, using Hologic QDR-4500W instrumentation (Hologic Corporation, Waltham, Massachusetts).

### Assessment of regional LBM and FBM by dual x-ray absorptiometry

DXA measurements related to whole body DXA scans were extracted from Apex Software version 3.4.

The densitometric image of each patient was divided, following the manufacturer's instructions, into different body districts, including arms, legs, trunk, head, and other derived regions, such as the android and gynoid zone.

BMD, bone mineral content (BMC), fat free mass, and fat mass were assessed for every region of interest, where fat free mass was provided by the software in terms of lean soft tissue plus BMC. Despite the lean mass measured by DXA counts also skin, connective tissue, and some lean components within the adipose tissue (*Visser et al., 1999*), it still correlates highly with computed tomography and magnetic resonance imaging measurements and represents a good approximation of the real muscle mass (*Buckinx et al., 2018*).

Other DXA-derived body composition parameters, such as fat mass index (FM/H2) (FMI), appendicular lean mass index (ALM/H2) (ALMI), and trunk/appendicular ratio, were then calculated to complete the analysis and the patient characterization.

## Circulating bone turnover markers

Blood chemistry and bone turnover markers: alkaline phosphatase (ALP) and C-terminal telopeptide of type I collagen (CTX) were assessed at baseline, 6 and 12 months. CTX serum levels were measured using the ElectroChemiLuminescenceAssay (ECLIA) kit Elecsys beta-CrossLaps/serum (Roche Diagnostic, Germany) using Cobas e411instruments (Roche); normal ranges were <0.704 ng/ml (men of the age between 50 and 70), <0.854 ng/ml (men >70) with a repeatability (coefficient of variation) CV% of 2.6. Bone ALP serum levels were determined in a two-step procedure. Briefly, total ALP serum activity was measured using the colorimetric method ALP2 (Roche) using Cobas c701 instruments (Roche); normal ranges were 50–116 U/l ± 0.6 with a repeatability CV% of 0.7. Samples were then subjected to electrophoretic separation to separate the different ALP isoforms using the G26 automated system (Sebia, France) equipped with the Interlab specific kit (Italy). Bone ALP activity was calculated as fraction of the total ALP activity related to the percentage of densitometric analysis of the electrophoretic migration.

## Statistical analysis

The dataset is available in supplementary material (*Figure 1—source data 1*). The normal distribution of continuous variables was tested with the Shapiro–Wilk test. Differences between parameters at baseline and 12 months were computed as percentage changes. To test if these changes were significantly different from 0 we used one sample *t*-test, or alternatively the non-parametric sign test.

Correlations between variables either at baseline and 12 months were expressed as Pearson's *r* (or alternatively with Spearman *R*, for variables not normally distributed).

We considered a significant threshold of p < 0.05, and to control for possible false positive results we applied the Bonferroni correction ($p' = \frac{p}{k}$, where *k* is the number of hypotheses tested). Given that type I errors cannot decrease (the whole point of Bonferroni adjustments) without inflating type II errors (*Perneger, 1998*), significant results in the raw test but which do not remain so after correction will still be mentioned.

Our post hoc power analysis, conducted with 29 participants, indicates that our study is powered at 80% to detect a correlation of at least 0.502.

Statistics were performed using R and SPSS (IBM Corp. Released 2015. IBM SPSS Statistics for Windows, Version 23.0. Armonk, NY: IBM Corp.).

## Results

Twenty-nine patients were included in the BLADE study and their characteristics have been described elsewhere (*Palumbo et al., 2022*). The consort diagram is reported in supplementary material (*Figure 1—figure supplement 1*). An outlier value of FSH after 12 months was removed from the analysis. Mean FSH serum levels at baseline conditions were 11.9 UI/l (95% confidence intervals [CIs]: 7.6–16.3). Mean FSH serum levels significantly decreased to 1.45 UI/l (95% CIs: 1.01–1.89) after 6 months of degarelix treatment and to 2.4 UI/l (95% CIs: 1.4–3.4) after 12 months. The corresponding percent changes were −77.59% (95% CIs: −86.20 to −68.87) and −59.7% (95% CIs: −80.3 to −39.1, p < 0.001) (*Figure 1*). Data on changes in body composition parameters, BMD, and bone turnover markers have been published elsewhere (*Palumbo et al., 2021*; *Palumbo et al., 2022*; *Dalla Volta et al., 2024*). Correlations between FSH changes and changes in body composition parameters, BMD, and bone turnover markers were not statistically significant, so these data are not presented.

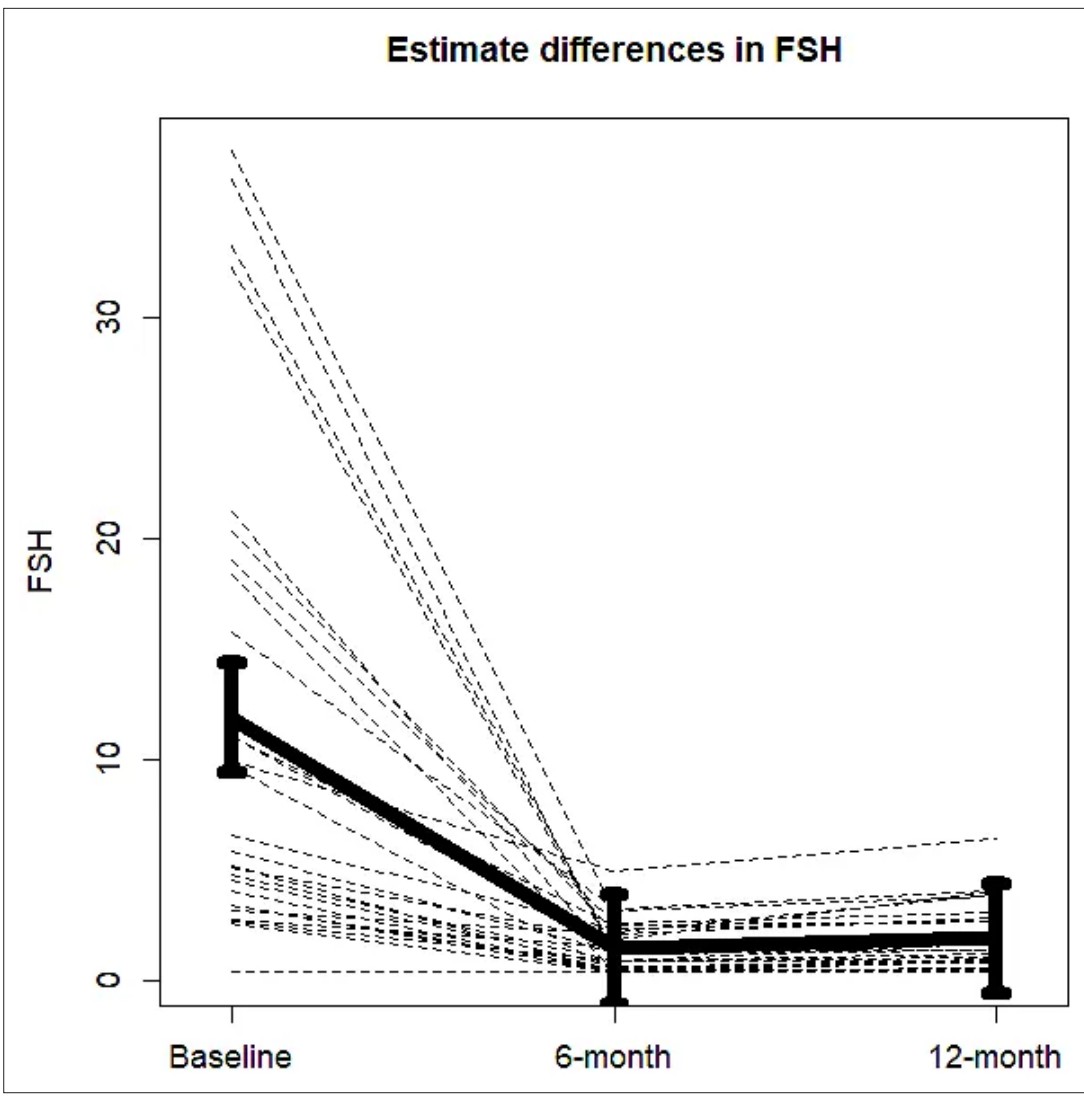

**Figure 1.** Changes in mean follicle-stimulating hormone (FSH) serum levels from baseline to 6 and 12 months of degarelix treatment.

The online version of this article includes the following source data and figure supplement(s) for figure 1:

**Source data 1.** Data are from BLADE study database.

**Figure supplement 1.** Consort diagram.

## Relationship between FSH serum levels and body composition, BMD, and bone turnover markers at baseline

At baseline conditions no relationship was found between FSH serum levels and the following DXA-derived parameters: BMD at left hip and spine, total FBM, total LBM, lean mass at arms and legs, ALMI, ALMI/FBM ratio, and android/gynoid ratio. No correlation was also found between FSH serum levels and serum levels of either ALP or CTX. A direct relationship, although not significant, was found between FSH serum levels and FBM at arms ($r = 0.36$) and legs ($r = 0.33$) (*Table 1*).

## Relationship between FSH serum levels and body composition, BMD, and bone turnover markers after 12 months of degarelix treatment

The correlation between circulating FSH levels at 6 and 12 months, bone turnover markers evaluated at the same time, BMD and body composition parameters assessed after 12 months is shown in *Table 2*. FSH serum levels assessed after 6 months of degarelix administration showed a direct

**Table 1.** Baseline relationships between FSH serum levels and BMD, bone turnover markers, and body composition parameters.

**FSH**

|  | Correlation coefficient | p |
|---|---|---|
| TOT BMD | 0.240 | 0.218 |
| BMD LEFT HIP | 0.229 | 0.240 |
| BMD L2–L4 | 0.148 | 0.453 |
| CTX | −0.115 | 0.560 |
| BONE ALP | 0.001 | 0.997 |
| TOT fat (g) | 0.274 | 0.158 |
| Arm fat (g) | 0.363 | 0.058 |
| Leg fat (g) | 0.330 | 0.087 |
| Head fat (g) | 0.228 | 0.244 |
| Trunk fat (g) | 0.249 | 0.202 |
| TOT lean (g) | 0.216 | 0.270 |
| Arm lean (g) | 0.166 | 0.400 |
| Leg lean (g) | 0.197 | 0.315 |
| Head lean (g) | 0.250 | 0.199 |
| Trunk lean (g) | 0.190 | 0.333 |
| ALMI (appendicular lean/Ht$^2$) | 0.269 | 0.167 |
| ALMI/FBM | −0.184 | 0.349 |
| Android/gynoid ratio | −0.024 | 0.902 |

Data are Spearman R.
FSH: follicle-stimulating hormone. TOT: total. BMD: bone mineral density. CTX: C-terminal telopeptide of type I collagen. ALP: alkaline phosphatase. ALMI: appendicular lean mass index. Ht$^2$: height squared. FBM: fat body mass.

relationship with total FBM ($r = 0.49$, p = 0.008) and with fat mass evaluated at arms ($r = 0.53$, p = 0.004), trunk ($r = 0.48$, p = 0.009), and legs ($r = 0.45$, p = 0.015). In contrast, FSH serum levels after 6 months of treatment showed an inverse relationship with ALMI/FBM ratio ($r = −0.58$, p = 0.001). FSH measured after 12 months maintained a significant relationship with total FBM ($r = 0.52$, p = 0.006) (*Figure 2a*), fat mass at arms ($r = 0.54$, p = 0.004) (*Figure 2b*), and fat mass at trunk ($r = 0.45$, p = 0.018) (*Figure 2c*). The relationship between FSH and fat mass at legs lost the statistical significance ($r = 0.33$, p = 0.089), while the inverse relationship with ALMI/FBM ratio was confirmed ($r = −0.64$, p = 0.001) (*Figure 2d*).

## Discussion

ADT in PC patients leads to an increase in fat mass and to a decrease in lean mass, thus increasing the risk of sarcopenic obesity. In addition, ADT has also a negative impact on skeletal fragility, which increases the risk of fractures. During the recent years, there has been consistent evidence that alterations in body composition could contribute to determine skeletal fragility in patients under ADT, such as well demonstrated in other forms of male hypogonadism (*Vena et al., 2023*). However, the determinants of altered body composition in male hypogonadism and in ADTs have not been completely defined. As a matter of fact, low testosterone values do not seem to be the only determinant of sarcopenic obesity in males with hypogonadism. In this context, FSH values might play a role. In fact, among its extragonadal effects, FSH directly affects adipogenesis, lean mass and bone turnover (*Liu et al., 2017*; *Rojekar et al., 2023*; *Zaidi et al., 2023*; *Gera et al., 2022*).

**Table 2.** Relationships between FSH serum levels and BMD, bone turnover markers, and body composition parameters after 12 months degarelix treatment.

| | FSH at 6 **months** | | FSH at 12 **months** | |
|---|---|---|---|---|
| | Correlation coefficient | p | Correlation coefficient | p |
| TOT BMD | −0.039 | 0.843 | −0.107 | 0.595 |
| BMD L2–L4 | 0.102 | 0.604 | −0.040 | 0.844 |
| BMD LEFT HIP | 0.030 | 0.881 | −0.044 | 0.829 |
| CTX | 0.051 | 0.796 | 0.055 | 0.785 |
| BONE ALP | 0.174 | 0.376 | 0.227 | 0.256 |
| TOT fat (g) | **0.490** | **0.008** | **0.518** | **0.006** |
| Arm fat (g) | **0.532** | **0.004** | **0.541** | **0.004** |
| Leg fat (g) | **0.454** | **0.015** | 0.328 | 0.089 |
| Head fat (g) | 0.187 | 0.341 | 0.093 | 0.644 |
| Trunk fat (g) | **0.483** | **0.009** | **0.452** | **0.018** |
| TOT lean (g) | 0.029 | 0.885 | −0.036 | 0.859 |
| Arm lean (g) | 0.041 | 0.837 | 0.028 | 0.889 |
| Leg lean (g) | −0.176 | 0.371 | −0.285 | 0.149 |
| Head lean (g) | 0.086 | 0.663 | −0.006 | 0.977 |
| Trunk lean (g) | 0.161 | 0.414 | 0.021 | 0.916 |
| ALMI (appendicular lean/Ht$^2$) | −0.229 | 0.241 | −0.265 | 0.181 |
| ALMI/FBM | **−0.581** | **0.001**[*] | **−0.604** | **0.001**[*] |
| Android/gynoid ratio | 0.142 | 0.472 | 0.231 | 0.246 |

Data are Spearman *R*.

Bold values are those attaining statistical significance.

[*]Still significant after Bonferroni correction.

FSH: follicle-stimulating hormone. TOT: total. BMD: bone mineral density. CTX: C-terminal telopeptide of type I collagen. ALP: alkaline phosphatase. ALMI: appendicular lean mass index. Ht$^2$: height squared. FBM: fat body mass.

As demonstrated in other studies prospectively evaluating circulating FSH levels after treatment with LHRH agonists and antagonists (*Klotz et al., 2008*), degarelix administration consistently reduced FSH values, although values might be variable among individuals under ADT. In this context, it is reasonable to hypothesize that variable values of FSH might directly influence body composition and skeletal fragility.

This ancillary analysis of the BLADE study shows that higher serum levels of FSH during degarelix treatment were significantly associated with higher total, limbs, and trunk fat mass. Moreover, higher FSH values under ADT were associated with lower ALMI/FBM ratio, a parameter that identifies patients at high risk for sarcopenic obesity. We can infer therefore that the effect was caused by the direct relationship of FBM since there was no correlation between lean mass and FSH in this study. These associations were not significant before starting degarelix, suggesting that sex hormones might counteract FSH effect on body composition at baseline conditions, whereas this effect becomes more evident after ADT. According to these data, small changes in FSH can result in significant changes in body composition in a condition of androgen and estrogen deprivation. However, since sex hormones were not measured in our patients, it was unclear whether the possible variability in testosterone values during degarelix treatment (*Kamada et al., 2024*) might have influenced the detrimental effects of FSH and body composition. Indeed, the protocol of BLADE study did not include measurement of testosterone values, since radioimmunoassay and chemiluminescence assay used in clinical practice

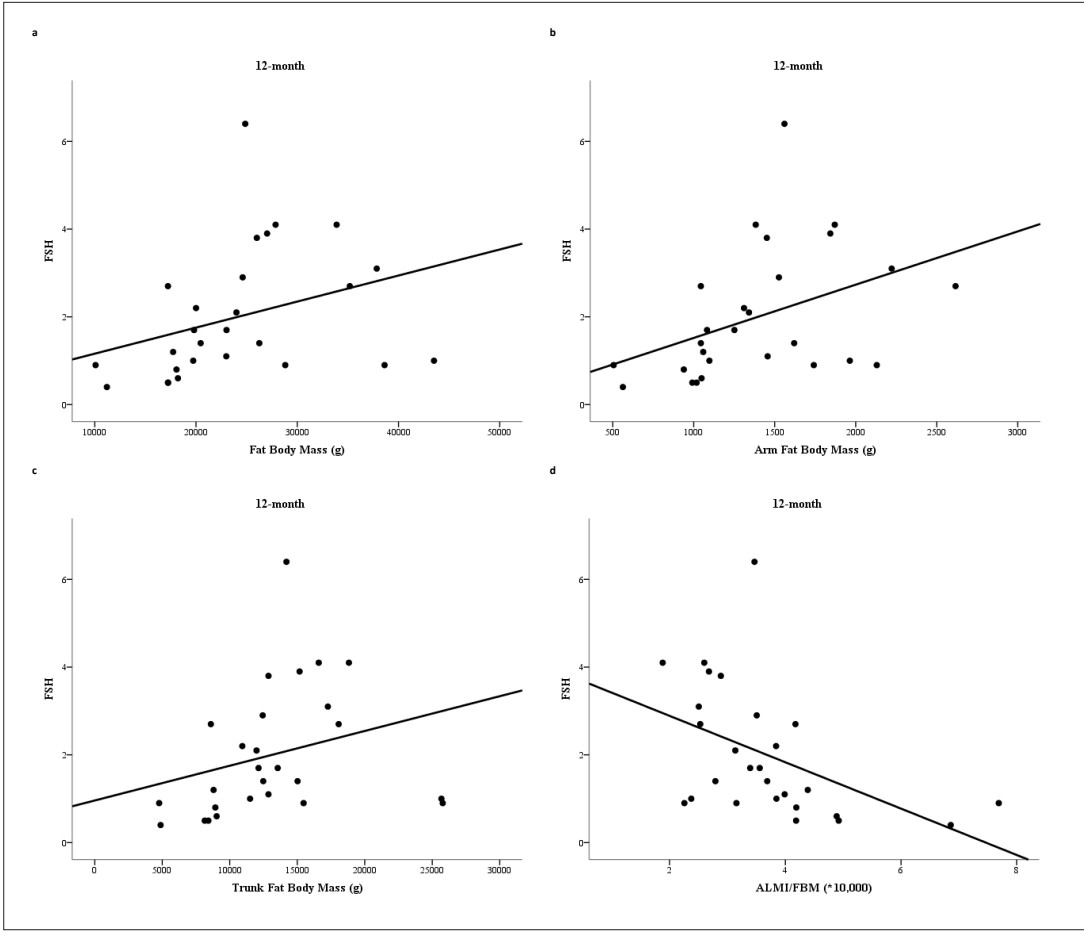

**Figure 2.** Relationship between follicle-stimulating hormone (FSH) serum levels and fat body mass (FBM) (**a**), arm FBM (**b**), trunk FBM (**c**), and appendicular lean mass index (ALMI)/FBM ratio (**d**) after 12 months degarelix treatment. *R* = Spearman correlation coefficient; p = p-value.

could overestimate testosterone levels in majority of patients under ADTs, leaving a concern of misdiagnosing truly castrate patients as being inadequately treated (*Tiwari et al., 2022*).

It is known that one of the most noticeable differences of LHRH antagonists compared to LHRH agonists are FSH levels during therapy, which are mildly higher with LHRH agonists compared to LHRH antagonists. According to our data, PC patients treated with LHRH antagonists may experience less body composition changes due to lower FSH residual values (*Klotz et al., 2008*; *Margel et al., 2019*; *Abufaraj et al., 2021*).

Unlike other settings, FSH values before and after degarelix did not correlate with either BMD or bone turnover markers.

To our knowledge, this is the first study showing an association between circulating FSH levels and body composition in PC patients undergoing ADT. However, this study suffers from all the limitations of ancillary studies. Therefore, the results obtained are to be considered as hypothesis generating and cannot be generalized.

In conclusion, the findings of this study suggest that FSH serum levels after ADT could impact body composition – FBM in particular – in PC patients undergoing ADT.

Of course further investigation is required to establish the association between FSH serum levels, measured during ADT, and sarcopenic obesity risk in PC patients. However, this is an area of great interest.

FSH could be a promising marker in monitoring PC patients during ADT in order to identify those who are more vulnerable and at greater risk of alterations in body composition, which can be captured and measured reliably by DXA (*Mazziotti et al., 2024*).

## Acknowledgements

The authors are grateful to the following associations: AOB (Associazione Oncologica Bresciana) and FIRM-Onlus (Fondazione Internazionale di Ricerca in Medicina Onlus) for the continuous support. This research was partially funded by Ferring Pharmaceuticals. The funder had no role in design and conduct of the study, collection, management, analysis, and interpretation of the data and in preparation, review, or approval of the manuscript.

## Additional information

### Competing interests

Luca Rinaudo: L Rinaudo is affiliated with Tecnologie Avanzate S.r.l. The author has no financial interests to declare. The other authors declare that no competing interests exist.

### Funding

| Funder | Grant reference number | Author |
| --- | --- | --- |
| Ferring | .. | Alfredo Berruti |

The funders had no role in study design, data collection, and interpretation, or the decision to submit the work for publication.

### Author contributions

Marco Bergamini, Data curation, Investigation, Methodology, Writing – original draft; Alberto Dalla Volta, Data curation, Investigation, Methodology, Writing – review and editing; Carlotta Palumbo, Stefania Zamboni, Luca Triggiani, Investigation, Methodology; Manuel Zamparini, Data curation, Formal analysis; Marta Laganà, Nunzia Di Meo, Irene Caramella, Francesca Valcamonico, Paolo Borghetti, Andrea Guerini, Alessandro Antonelli, Investigation; Luca Rinaudo, Validation, Investigation, Methodology; Roberto Bresciani, Conceptualization, Writing – review and editing; Davide Farina, Conceptualization, Investigation, Methodology, Writing – review and editing; Claudio Simeone, Conceptualization, Writing – original draft; Gherardo Mazziotti, Conceptualization, Supervision, Writing – review and editing; Alfredo Berruti, Conceptualization, Data curation, Supervision, Funding acquisition, Writing – review and editing

### Author ORCIDs

Marco Bergamini https://orcid.org/0000-0001-6161-707X
Alberto Dalla Volta https://orcid.org/0000-0001-8009-7663
Davide Farina https://orcid.org/0000-0003-4075-0001
Gherardo Mazziotti https://orcid.org/0000-0001-5458-701X

### Ethics

Clinical trial registration clinicalTrials.gov NCT03202381, EudraCT Number 2016-004210-10.
The study was carried out in accordance with the Declaration of Helsinki Principles and Good Clinical Practices and was approved by the Ethics Committee of Brescia (approval number NP2540).

### Decision letter and Author response

Decision letter https://doi.org/10.7554/eLife.92655.sa1
Author response https://doi.org/10.7554/eLife.92655.sa2

## Additional files

### Supplementary files

• Supplementary file 1. Clinical and demographical data of enrolled patients at baseline.

• MDAR checklist

• Reporting standard 1. STROBE statement for prospective study.

## Data availability

All data generated or analysed during this study are included in the manuscript. Source data have been provided in an excel file labelled Source data 1.

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
