## [Editor Report]

This ancillary study from the BLADE study provided valuable findings regarding FSH as a predictor of sarcopenic obesity in patients using the LHRH antagonist Degarelix, suggesting a potential direct effect of FSH on fat. The study design and analysis were solid, and the limitations of the study such as the small sample size and lack of sex hormone analysis-were properly addressed.

---

## [Decision Letter]

**Decision letter after peer review:**

Thank you for submitting your article "Relationship between circulating FSH levels and body composition and bone health in patients with prostate cancer who undergo androgen deprivation therapy: The BLADE study" for consideration by *eLife*. Your article has been reviewed by 2 peer reviewers, one of whom is a member of our Board of Reviewing Editors, and the evaluation has been overseen by Diane Harper as the Senior Editor.

Essential revisions (for the authors):

1. The relationship between FSH and body composition can be further elucidated by incorporating changes in testosterone levels and addressing other major confounding variables in the analysis.

2. The claim of sarcopenia and FSH, postulated by using ALBI/FBM as a variable, seems weak and requires additional supportive evidence.

*Reviewer #1 (Recommendations for the authors):*

If the authors can provide, the additional regression model might be helpful to understand the direct relationship between FSH and fat, especially sex hormones.

Please address the discrepancy of correlation coefficient (r) between FSH and ALBI in this manuscript and the earlier paper (Palumbo et al., Prostate Cancer Prostatic Dis 2021) -0.64, p = 0.001 (this manuscript) vs. -0.44 p = 0.02 (Palumbo et al., Prostate Cancer Prostatic Dis 2021).

*Reviewer #2 (Recommendations for the authors):*

A few points which should be addressed:

– A supplementary table with the general characteristics of patients should be provided, e.g. in a supplementary table, for a better understanding of the study results, instead of referring to the previous publication.

– To make the manuscript more understandable (even for non-experts in the field) a brief mention to the mechanism of action of Degarelix should be given.

– Given the limited number of patients recruited for the study, power calculations should be provided.

[Editors' note: further revisions were suggested prior to acceptance, as described below.]

Thank you for resubmitting your work entitled "Relationship between circulating FSH levels and body composition and bone health in patients with prostate cancer who undergo androgen deprivation therapy: The BLADE study" for further consideration by *eLife*. Your revised article has been evaluated by Diane Harper (Senior Editor) and a Reviewing Editor.

The manuscript has been improved but there are some remaining issues that need to be addressed, as outlined below:

The authors addressed all the questions arising from the reviewers.

However, I still have a couple of remaining questions and suggestions. Firstly, while the authors' claim regarding FSH as a predictive marker for sarcopenic obesity seems convincing, associating it with cardiovascular complications appears far-fetched and is not supported by any evidence from this study.

The sentences in the discussion (lines 209-211, page 8) are unclear. This section serves as a summary of their findings and should be conveyed more clearly to the readers.

If I'm not mistaken, there was no correlation between FSH and any of the body composition parameters derived from DEXA. However, with Degarelix treatment, FSH became positively and negatively correlated with fat mass and ALMI/FBM, respectively.

The authors acknowledged that sex hormones were not measured in this protocol and the challenge of measuring them in the setting of Degarelix treatment. This limitation needs to be relayed to the readers in the discussion. I believe many readers will question the lack of data, given the intricate axis and the powerful impact of sex hormones on body composition.

Please change "sexual hormones" (line 215, page 9) to "sex hormones".

---

## [Author Response]

Essential revisions (for the authors):Reviewer #1 (Recommendations for the authors):If the authors can provide, the additional regression model might be helpful to understand the direct relationship between FSH and fat, especially sex hormones.

Regrettably, this cannot be accomplished because we do not have the circulating levels of estrogen and testosterone.

Please address the discrepancy of correlation coefficient (r) between FSH and ALBI in this manuscript and the earlier paper (Palumbo et al., Prostate Cancer Prostatic Dis 2021) -0.64, p = 0.001 (this manuscript) vs. -0.44 p = 0.02 (Palumbo et al., Prostate Cancer Prostatic Dis 2021).

We realized that 1 patient had a very high FSH value at 12 months in contrast to all other patients, we decided to remove this outlier value. We specified this in the new manuscript (page 6, lines 22-23) and in the Consort diagram.

Reviewer #2 (Recommendations for the authors):A few points which should be addressed:– A supplementary table with the general characteristics of patients should be provided, e.g. in a supplementary table, for a better understanding of the study results, instead of referring to the previous publication.

As suggested by the reviewer, a table of patients characteristics was added in the supplementary materials.

– To make the manuscript more understandable (even for non-experts in the field) a brief mention to the mechanism of action of Degarelix should be given.

This was done in the new version: page 4, lines 7-8.

– Given the limited number of patients recruited for the study, power calculations should be provided.

This was done.

[Editors’ note: what follows is the authors’ response to the second round of review.]

The manuscript has been improved but there are some remaining issues that need to be addressed, as outlined below:The authors addressed all the questions arising from the reviewers.However, I still have a couple of remaining questions and suggestions. Firstly, while the authors' claim regarding FSH as a predictive marker for sarcopenic obesity seems convincing, associating it with cardiovascular complications appears far-fetched and is not supported by any evidence from this study.

We changed the conclusions in both the abstract (page 2, lines 22, 23) and the manuscript (page 9, lines 23-25).

The sentences in the discussion (lines 209-211, page 8) are unclear. This section serves as a summary of their findings and should be conveyed more clearly to the readers.If I'm not mistaken, there was no correlation between FSH and any of the body composition parameters derived from DEXA. However, with Degarelix treatment, FSH became positively and negatively correlated with fat mass and ALMI/FBM, respectively.

We reworded the above sentences (page 8, line 19-25; page 9, lines 1-2).

The authors acknowledged that sex hormones were not measured in this protocol and the challenge of measuring them in the setting of Degarelix treatment. This limitation needs to be relayed to the readers in the discussion. I believe many readers will question the lack of data, given the intricate axis and the powerful impact of sex hormones on body composition.

This limitation was more clearly aknowledged and discussed (page 9, lines 2-8).

Please change "sexual hormones" (line 215, page 9) to "sex hormones".

Done (page 9, line 2).